# TOWARDS LONG-FORM SPATIO-TEMPORAL VIDEO GROUNDING

## ABSTRACT

Videos can span several minutes or even hours in real scenarios, yet current research on spatio-temporal video grounding (STVG), given a textual query, mainly focuses on localizing target from a video of tens of seconds, typically less than one minute, limiting its applications. In this paper, we explore *Long-Form STVG* (**LF-STVG**), that aims to locate the target in long-term videos. In LF-STVG, long-term videos encompass a much longer temporal span and more irrelevant information, making it challenging for current short-form STVG models that process all frames at once. Addressing these, we introduce a novel ***Auto*R*egressive* *T*ransformer framework for LF-STVG* (**ART-STVG**). Unlike current STVG methods requiring seeing the entire sequence to make a full prediction at once, our ART-STVG treats the video as a streaming input and processes its frames sequentially, making it capable of easily handling the long videos. To capture spatio-temporal context in ART-STVG, spatial and temporal memory banks are developed and applied to decoders of ART-STVG. Considering that memories at different moments are not always relevant for localizing the target in current frame, we propose simple yet effective memory selective strategies that enable more relevant information for the decoders, greatly improving performance. Moreover, rather than parallelizing spatial and temporal localization as done in existing approaches, we introduce a novel cascaded spatio-temporal design that connects spatial decoder to temporal decoder during grounding. This way, our ART-STVG leverages more fine-grained target information to assist with complicated temporal localization in complex long videos, further boosting the performance. On the newly extended datasets for LF-STVG, ART-STVG largely outperforms current state-of-the-art approaches, while showing competitive results on conventional Short-Form STVG. Our code and models will be released.

## 1 INTRODUCTION

Spatio-temporal video grounding (**STVG**) aims at localizing the target of interest in *space* and *time* from an untrimmed video given a *free-form* textual query (Zhang et al., 2020b). As a multimodal task, it needs to accurately comprehend spatio-temporal content of a video and make connections to the provided textual query for target localization. Owing to its important role in multimodal video understanding, STVG has recently attracted extensive attention (Zhang et al., 2020b; Jin et al., 2022a; Su et al., 2021b; Tang et al., 2021; Yang et al., 2022; Zhang et al., 2020a; Lin et al., 2023b; Gu et al., 2024; Wasim et al., 2024; Gu et al., 2025).

Despite advancements, existing research mainly focuses on locating the desired target from a short-term video of tens of seconds, typically *less than* one minute. For instance, the average video length of existing popular datasets HCSTVG-v1/-v2 (Tang et al., 2021) and VidSTG (Zhang et al., 2020b) is 20 and 35 seconds, respectively. Nonetheless, in real-world applications, such as video retrieval and visual surveillance, the videos can span several *minutes* or even *hours*, which results in a large *gap* between current research (focusing on target localization from *short-term* videos) and practical applications (the need of target localization in *long-term* videos). To mitigate this gap, we explore *Long-Form STVG* (**LF-STVG**), which locates the target of interest in *long-term* videos given a query.

To localize desired target, current STVG methods (Wasim et al., 2024; Jin et al., 2022a; Lin et al., 2023b; Yang et al., 2022; Gu et al., 2024; 2025) process *all* the video frames in one time (see Fig. 1 (a)), aiming at capturing and leveraging global context from the entire video for localization. These

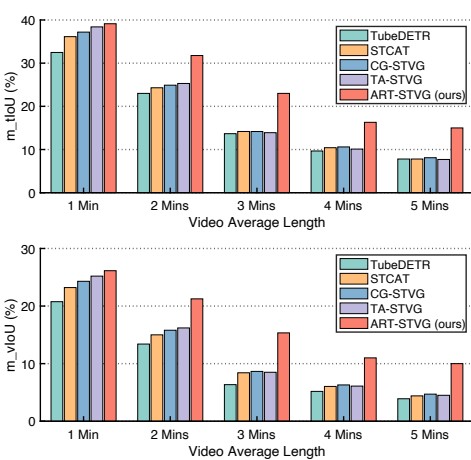

(a) Existing STVG methods: *processing all frames in one time*

(b) Our proposed ART-STVG: *processing one frame at a time*

Figure 1: Comparison between existing STVG approaches (Yang et al., 2022; Gu et al., 2024; Jin et al., 2022a; Lin et al., 2023b; Wasim et al., 2024; Gu et al., 2025) that see the entire video sequence to make a full prediction at once in (a) and our ART-STVG that ingests frames one at a time and hence is suitable for LF-STVG in (b). *Best viewed in color for all figures.*

approaches have achieved impressive results on Short-Form STVG (SF-STVG). Nonetheless, as the videos grow longer, new challenges arise, leading us to a critical question: ***Is this way of processing all frames in one time for current SF-STVG applicable to LF-STVG?*** Our answer is ***negative***! In LF-STVG, videos often encompass longer temporal span, which largely increases the complexities of spatio-temporal localization. In addition, long videos commonly contain far more irrelevant information, requiring the model to identify the target event from extensive redundant content. For these reasons, processing all frames of a long video at once, as done in current STVG methods, presents significant challenges in capturing long-term spatio-temporal relationships and handling excessive irrelevant information for accurate localization (see Fig. 2). Additionally, it causes computational bottlenecks because of high GPU memory requirements for simultaneous feature learning and target localization in all video frames.

Addressing the aforementioned challenges, we propose a novel **A**uto**R**egressive **T**ransformer method for LF-**STVG**, dubbed ***ART-STVG***. Specifically, it treats the video as a streaming input and processes its frames sequentially (see Fig. 1 (b)). To capture the

Figure 2: Comparison of current STVG methods and our ART-STVG on different LF-STVG benchmarks. We can see that, ART-STVG significantly surpasses existing models for target localization in long videos. Furthermore, we observe that, the longer the video is, the more significant the improvement of ART-STVG over other methods is.

crucial spatio-temporal contextual information in videos, we maintain two memory banks, that reserve essential spatio-temporal information from videos, for spatial and temporal decoders in ART-STVG. Since the memories in the bank are *not* equally important to a certain frame, we introduce simple yet effective memory selective strategies to leverage more relevant information in memory banks for grounding, effectively boosting performance. Compared to existing approaches which require seeing the entire video for prediction, our proposed ART-STVG ingests frames one at a time for prediction, hence naturally processing longer videos and resolving the computational bottleneck faced by current approaches. Furthermore, rather than parallelizing the spatial and temporal localization as is done in existing approaches, we propose a novel cascaded spatio-temporal design which connects spatial decoder to temporal decoder during grounding. By doing so, ART-STVG is able to enjoy more fine-grained target information from the spatial decoder to assist with the more complicated temporal localization, further boosting performance. Fig. 3 shows the architecture of ART-STVG. To our best knowledge, this paper is the *first* to explore the LF-STVG problem, and our ART-STVG is the *first* framework attempting to handle LF-STVG.

To verify the effectiveness of our ART-STVG, we extend validation set of the short-term benchmark HCSTVG-v2 (Tang et al., 2021) (the reason for choosing HCSTVG-v2 for extension is described later). Specifically, we extend its average video length from 20 seconds to 1∼5 minutes, hence referred

to as LF-STVG-1min/2min/3min/4min/5min. We conduct extensive experiments on both long-form and short-form STVG. The results show that, ART-STVG outperforms all existing approaches on LF-STVG by achieving new state-of-the-arts, while showing competitive performance on SF-STVG.

In summary, our **contributions** are as follows: ♠ We introduce a novel memory-augmented autoregressive transformer, dubbed ART-STVG, for LF-STVG; ♥ We design memory selection strategies that allow the selection of relevant crucial spatio-temporal context for enhancing target localization; ♣ We propose a cascaded spatio-temporal decoder design to fully utilize the fine-grained information produced by spatial localization to assist temporal localization; ♦ In our extensive experiments on both long-term and short-term benchmarks, our ART-STVG achieves excellent performance.

## 2 RELATED WORK

**Spatio-temporal video grounding (STVG)** aims to localize a spatial-temporal tube in an untrimmed video that corresponds to the given text query. Early methods (Tan et al., 2021; Yu et al., 2021; Wang et al., 2022; Zhang et al., 2020b; Su et al., 2021a) are predominantly two-stage approaches. These approaches first adopt a pre-trained object detector (Ren et al., 2015) to generate object proposals, and then select the proposals based on the given textual query. Such methods are easily limited by the pre-trained object detector. Recent approaches (Jin et al., 2022a; Lin et al., 2023b; Talal Wasim et al., 2024; Gu et al., 2024; 2025), inspired by DETR (Carion et al., 2020), propose one-stage frameworks that directly generate tubes for target localization, displaying better performance than the two-stage models. Nevertheless, both the early two-stage and recent one-stage approaches focus on SF-STVG and process the entire video at one time for simultaneous target localization in all frames. **Different from** existing methods, our ART-STVG is specially designed for LF-STVG. Specifically, ART-STVG treats the video as a streaming input and processes its frames sequentially with an autoregressive framework, thus making it more suitable for handling long-term video sequences.

**Long-term video understanding** has been explored in many tasks such as action detection (Cheng & Bertasius, 2022), video captioning (Islam et al., 2024), and video question answering (Song et al., 2024; Cheng et al., 2024; He et al., 2024). Its main challenge is that capturing complex spatio-temporal dependencies over long durations requires high computational cost. To address this, early methods (Donahue et al., 2015; Wu & Krahenbuhl, 2021) model pre-extracted video features without jointly training the backbone. Recent works (Bai et al., 2023; Zhang et al., 2024) design efficient strategies to process more frames simultaneously, while others (Wu et al., 2022; He et al., 2024; Qian et al., 2025; Wang et al., 2024) construct streamlined transformers with memory banks for video understanding. **Different from** these works, we focus on long-term STVG. Besides, **unlike** memory banks in video question answering (Song et al., 2024; He et al., 2024) for global context learning, the memory in ART-STVG aims to capture text-guided spatial instance and temporal event boundary cues, which, together with our memory selection, are specially designed for LF-STVG.

**Autoregressive architecture** has been studied and applied in various domains. Early autoregressive models are mainly based on recurrent neural networks (Medsker et al., 2001; Graves & Graves, 2012; Hochreiter & Schmidhuber, 1997). Recently, autoregressive transformer models (Vaswani et al., 2017; Katharopoulos et al., 2020; Touvron et al., 2023; Liu et al., 2024; Ren et al., 2024; Lin et al., 2023a) with attention mechanism have further advanced the field by enabling serial computation and capturing long-range dependencies. **Different from** these methods, we introduce an autoregressive transformer framework specially designed for LF-STVG.

## 3 THE PROPOSED APPROACH

**Overview.** We propose ART-STVG, a memory-augmented autoregressive transformer for LF-STVG. As shown in Fig. 3, the framework begins with a multimodal encoder (Sec. 3.1) that extracts and fuses visual and textual features. Following this, the cascaded spatio-temporal decoder performs autoregressive decoding for grounding (Sec. 3.2). Specifically, the memory-augmented spatial decoder (Sec. 3.3) captures the spatial location information of the target, while the memory-augmented temporal decoder (Sec. 3.4) focuses on learning the temporal location information.

Since ART-STVG processes frames sequentially, in the following description of our approach, we take the processing of the $i^{\text{th}}$ frame as an example for illustrating ART-STVG.

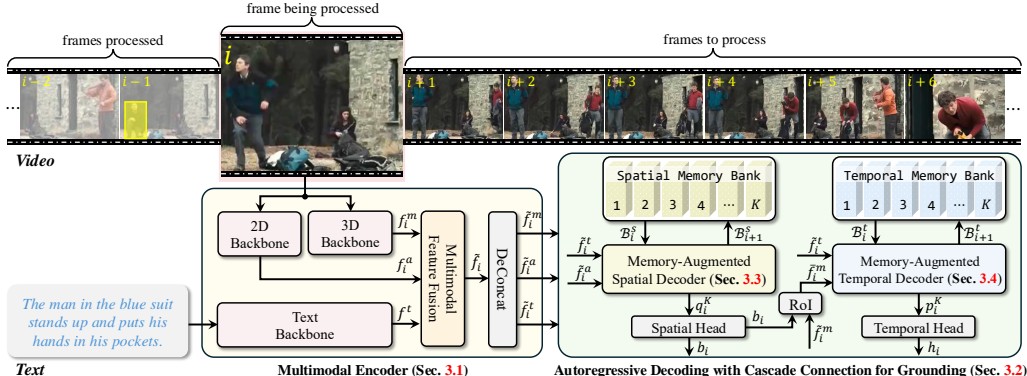

Figure 3: Architecture of our proposed ART-STVG, which comprises a multimodal encoder and an autoregressive decoder for target localization frame by frame.

## 3.1 MULTIMODAL ENCODER

Given video frame $i$ and the text, the multimodal encoder generates a multimodal feature, which is sent to the decoder for localization. It comprises feature extraction and fusion, as described below.

**Feature Extraction.** For the $i^{\text{th}}$ video frame, we extract its 2D appearance and 3D motion features to leverage rich static and dynamic cues. Specifically, the appearance feature is extracted using ResNet-101 (He et al., 2016), and the motion feature is extracted via VidSwin (Liu et al., 2022). Please **_note_**, when applying VidSwin to extract motion features, previous frames are also used as input. The appearance feature of frame $i$ is denoted as $f_i^a \in \mathbb{R}^{H \times W \times C_a}$, where $H$, $W$, and $C_a$ are height, width, and channel dimensions. Similarly, the motion feature is represented as $f_i^m \in \mathbb{R}^{H \times W \times C_m}$ with $C_m$ the channel dimension. For the text, we first tokenize it to a word sequence, and then apply RoBERTa (Liu et al., 2019) to extract its feature $f^t \in \mathbb{R}^{N_t \times C_t}$, where $N_t$ is the text feature length and $C_t$ the channel dimension.

**Feature Fusion.** Different modalities typically contain complementary information. Therefore, we fuse the appearance feature $f_i^a$ and motion feature $f_i^m$ of the $i^{\text{th}}$ video frame with the textual feature $f^t$ to generate a multimodal feature of the $i^{\text{th}}$ frame. Specifically, we first project them to the same channel dimension $C$, and then concatenate them to produce the multimodal feature $f_i^{'}$, as follows,

$$f_i^{'} = [\underbrace{f_{i_1}^a, f_{i_2}^a, ..., f_{i_{H \times W}}^a}_{\text{appearance feature } f_i^a}, \underbrace{f_{i_1}^m, f_{i_2}^m, ..., f_{i_{H \times W}}^m}_{\text{motion feature } f_i^m}, \underbrace{f_1^t, f_2^t, ..., f_{N_t}^t}_{\text{textual feature } f^t}] \tag{1}$$

Then, we adopt a self-attention encoder (Vaswani et al., 2017) to fuse multimodal features as follows,

$$\tilde{f}_i = \texttt{SelfAttEncoder}(f_i^{'} + \mathcal{E}_{pos} + \mathcal{E}_{typ}) \tag{2}$$

where $\mathcal{E}_{pos}$ and $\mathcal{E}_{typ}$ denote position and type embeddings, and $\texttt{SelfAttEncoder}(\cdot)$ is the self-attention encoder with $N$ ($N$=6) standard self-attention encoder blocks as in (Gu et al., 2024).

After obtaining $\tilde{f}_i$, we deconcatenate it to generate enhanced appearance, motion, and textual features $\tilde{f}_i^a$, $\tilde{f}_i^m$, and $\tilde{f}_i^t$ via $[\tilde{f}_i^a, \tilde{f}_i^m, \tilde{f}_i^t] = \texttt{DeConcat}(\tilde{f}_i)$ and apply them in decoder for target localization.

## 3.2 AUTOREGRESSIVE DECODING FOR GROUNDING

Our ART-STVG autoregressively decodes video frames to sequentially predict spatial and temporal target positions. As shown in Fig. 3, the decoding process of ART-STVG contains two parts, including spatial grounding and temporal grounding via two decoders. The former is responsible for predicting the spatial location of the target object, while the latter generates the temporal location of the target event. To capture spatio-temporal context in ART-STVG, spatial and temporal memory banks storing historical information, with effective memory selection, are developed and applied in the grounding process, largely enhancing performance. Besides, *rather than paralleling* the spatial and temporal grounding as done in current methods, we propose a novel **_cascaded_** design to connect spatial and temporal grounding in ART-STVG (see decoding part in Fig. 3). Such cascaded spatio-temporal

design allows ART-STVG to employ more fine-grained target cues from spatial grounding to assist with temporal localization in complex long videos, further improving ART-STVG for LF-STVG.

**Spatial Grounding.** In ART-STVG, the spatial grounding is achieved by learning a spatial query via iterative interaction with the multimodal feature. Let $q_i^0$ be the initial spatial query in the $i^{\text{th}}$ frame and $\mathcal{B}_i^s$ is the spatial memory bank at this moment. Given appearance feature $\tilde{f}_i^a$ and textual feature $\tilde{f}_i^t$ from $\tilde{f}_i$ in frame $i$, the interaction of spatial query with multimodal feature is achieved as follows,

$$q_i^K, \mathcal{B}_{i+1}^s = \texttt{MA-SpatialDecoder}(q_i^0, \mathcal{B}_i^s, [\tilde{f}_i^a, \tilde{f}_i^t]) \tag{3}$$

where $\texttt{MA-SpatialDecoder}(\cdot)$ is the memory-augmented spatial decoder with $K$ spatial decoder blocks (described in Sec. 3.3). It is worth ***noting*** that, the spatial memory bank $\mathcal{B}_i^s$ contains $K$ (*i.e.*, the number of decoder blocks) partitions, with each partition corresponding to a spatial decoder block. $q_i^K$ represents the final spatial query feature after $K$ decoder blocks, and $\mathcal{B}_{i+1}^s$ the new memory bank updated with spatial information from frame $i$ (see Sec. 3.3). After this, a spatial head, containing an MLP module, is used to predict the final object box $b_i$, as follows,

$$b_i = \texttt{SpatialHead}(q_i^K) \tag{4}$$

where $b_i \in \mathbb{R}^4$ is the central position, width, and height of the predicted target box in the $i^{\text{th}}$ frame.

**Temporal Grounding.** For temporal grounding, we learn a temporal query by interacting with the multimodal feature. To exploit the fine-grained spatial target cue to assist with temporal grounding, we design a cascade architecture. Specifically, with target box $b_i$ from spatial grounding, we first extract fine-grained target motion feature $\bar{f}_i^m \in \mathbb{R}^{1 \times 1 \times C}$ using RoI pooling (Ren et al., 2015) via

$$\bar{f}_i^m = \texttt{RoI}(\tilde{f}_i^m, b_i) \tag{5}$$

Compared to $\tilde{f}_i^m$, $\bar{f}_i^m$ is focused more on the target region and thus beneficial for localization.

After this, we interact the temporal query with multimodal feature. Let $p_i^0$ be the initial temporal query in frame $i$ and $\mathcal{B}_i^t$ the temporal memory bank at this moment. With fine-grained motion feature $\bar{f}_i^m$ and textual feature $\tilde{f}_i^t$, the interaction of temporal query and multimodal feature is performed via

$$p_i^K, \mathcal{B}_{i+1}^t = \texttt{MA-TemporalDecoder}(p_i^0, \mathcal{B}_i^t, [\bar{f}_i^m, \tilde{f}_i^t]) \tag{6}$$

where $\texttt{MA-TemporalDecoder}(\cdot)$ is the memory-augmented temporal decoder with $K$ temporal decoder blocks (described in Sec. 3.4). Similar to $\mathcal{B}_i^s$, the temporal memory bank $\mathcal{B}_i^t$ also comprises $K$ partitions, with each corresponding to a temporal decoder block. $p_i^K$ is the final temporal query feature after the decoder, and $\mathcal{B}_{i+1}^t$ the new memory bank updated with temporal information in frame $i$ (see Sec. 3.4). After this, a temporal head implemented with an MLP module is adopted for temporal localization in frame $i$, as follows,

$$h_i = \texttt{TemporalHead}(p_i^K) \tag{7}$$

where $h_i \in \mathbb{R}^2$ represents the event start probabilities $h_i^s$ and end probabilities $h_i^e$ of the $i^{\text{th}}$ frame.

By sequentially performing spatial and temporal grounding, we achieve target localization in each frame $i$, and meanwhile use information in frame $i$ to update memory banks for the next frame ($i$+1).

## 3.3 MEMORY-AUGMENTED SPATIAL DECODER

We propose a memory-augmented spatial decoder, guided by spatial memory from the spatial memory bank, to learn the target spatial position from the multimodal feature. Specifically, the memory-augmented spatial decoder comprises $K$ decoder blocks in a cascade for spatial grounding. As shown in Fig. 4 (a), each spatial decoder block corresponds to a partition in the spatial memory and contains two cross-attention blocks (Vaswani et al., 2017). Concretely, in the $k^{\text{th}}$ ($1 \leq k \leq K$) spatial decoder block, given the appearance feature $\tilde{f}_i^a$, the textual feature $\tilde{f}_i^t$, and the spatial query $q_i^{k-1}$ ($q_i^0$ initialized by zeros) of the $i^{\text{th}}$ frame, we first perform memory selection and then apply the selected memory to enhance the spatial query feature in spatial decoding.

**Spatial Memory Selection.** Since the spatial query contains crucial target information, we first insert the spatial query $q_i^{k-1}$ into the $k^{\text{th}}$ partition of spatial memory bank $\mathcal{B}_{i,k}^s$ corresponding to the $k^{\text{th}}$

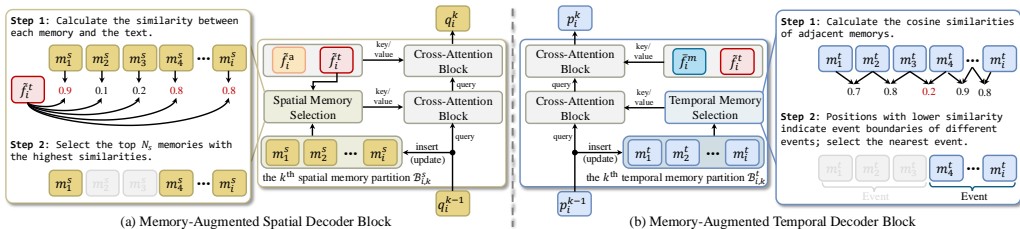

(a) Memory-Augmented Spatial Decoder Block (b) Memory-Augmented Temporal Decoder Block

Figure 4: The architectures of memory-augmented spatial and temporal decoder blocks in (a) and (b).

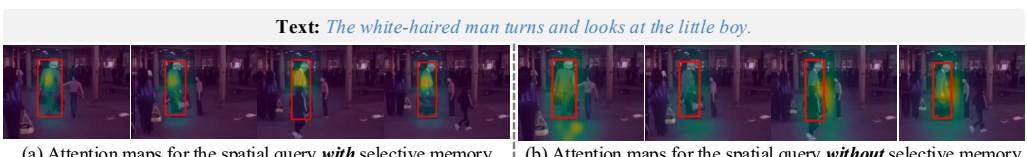

(a) Attention maps for the spatial query *with* selective memory (b) Attention maps for the spatial query *without* selective memory

Figure 5: Comparison of attention maps for spatial query *with* (in (a)) and *without* (in (b)) using selective spatial memory. The red box indicates the foreground target. We can see the use of selective spatial memory helps the model focus more on target regions, benefiting final target localization.

decoder block. Please **note** that, this insertion procedure also completes the update of each partition $\mathcal{B}_{i,k}^s$ in $\mathcal{B}_i^s$ to $\mathcal{B}_{i+1,k}^s$ in $\mathcal{B}_{i+1}^s$. Or in other words, we **update** the memory bank by simply adding the query as a new memory, without removing any existing memories.

After this, we perform memory selection from $\mathcal{B}_{i+1,k}^s$ for decoder block $k$. The **motivation** behind this selection is, the memories at different moments are not always relevant for target localization in current frame, and selecting more relevant information in spatial decoding enables learning better query feature for grounding. Specifically, the selective spatial memory $\mathcal{M}_{i,k}^s$ for block $k$ can be obtained via two steps in memory selection: *first*, we calculate the similarity between each spatial memory and the textual feature; *second*, based on similarity scores, the top $N_s$ spatial memories with the highest scores are selected to form $\mathcal{M}_{i,k}^s$. Fig. 4 (a) shows this spatial memory selection process.

**Memory-Augmented Spatial Decoding.** During decoding, we send $q_i^{k-1}$ to decoder block $k$ for learning $q_i^k$. To exploit spatial context, we first interact the query with selective spatial memory through a cross-attention block, as follows,

$$\tilde{q}_i^{k-1} = \texttt{CrossAtt}(q_i^{k-1}, \mathcal{M}_{i,k}^s) \tag{8}$$

where $\tilde{q}_i^{k-1}$ is the memory-augmented query feature in decoder block $k$, and $\texttt{CrossAtt}(\mathbf{u}, \mathbf{v})$ the cross-attention block (Vaswani et al., 2017), with $\mathbf{u}$ generating query and $\mathbf{v}$ key/value. After this, we further interact $\tilde{q}_i^{k-1}$ with the multimodal appearance and textual features for learning $q_i^k$, as follows,

$$q_i^k = \texttt{CrossAtt}(\tilde{q}_i^{k-1}, [\bar{f}_i^a, \tilde{f}_i^t]) \tag{9}$$

where $q_i^k$ is the learned query feature, and sent to next decoder block for further query feature learning.

Fig. 5 demonstrates the attention maps of spatial query with (see Fig. 5 (a)) and without (see Fig. 5 (b)) using selective spatial memory. We can clearly see using selective spatial memory helps the model focus more on target regions for better grounding. After $K$ spatial decoder blocks, the final spatial query feature $q_i^K$ is adopted for spatial prediction.

### 3.4 MEMORY-AUGMENTED TEMPORAL DECODER

The memory-augmented temporal decoder learns target temporal position using temporal memory from temporal memory bank. It has $K$ blocks for temporal grounding, with each corresponding to a temporal memory partition and containing two cross-attention blocks, as in Fig. 4 (b). In temporal decoder block $k$ ($1 \leq k \leq K$), given motion and textual features $\bar{f}_i^m$ and $\tilde{f}_i^t$, and temporal query $p_i^{k-1}$ ($p_i^0$ initialized by zeros), we first perform temporal memory selection and then apply the selected memory to enhance temporal decoding.

Figure 6: Illustration of selective temporal memory. In the middle figure, the attention sequence indicates cosine similarities of adjacent memories. Lower similarity (greener color) indicates potential event boundaries. Besides, in the up figure, we show predicted start and end probabilities, which are accurate to capture target event. The red box denotes the ground truth corresponding to the text query.

**Temporal Memory Selection.** The temporal query $p_i^{k-1}$ contains temporal event information. Thus, we first insert it into the $k^{th}$ partition of temporal memory bank $\mathcal{B}_{i,k}^t$ (also updating $\mathcal{B}_{i,k}^t$ to $\mathcal{B}_{i+1,k}^t$). Since long-term videos often contain multiple events, selecting relevant temporal memory related to the current event helps the temporal decoding better locate the event boundaries. To achieve this and obtain selective temporal memory $\mathcal{M}_{i,k}^t$, inspired by TextTiling (Hearst, 1997), we perform two steps in temporal memory section: in the *first* step, we calculate the similarities between the memories of adjacent frames; in the *second* step, points with lower similarities are considered as event boundaries between different events, and we only select memories corresponding to the event closest to current frame, as shown in Fig. 4 (b).

**Memory-Augmented Temporal Decoding.** In decoding, we send the temporal query $p_i^{k-1}$ to temporal decoder block $k$ for learning $p_i^k$. To exploit temporal context for enhancing query learning, we first interact the query with the selective temporal memory $\mathcal{M}_{i,k}^t$ by a cross attention block via

$$\tilde{p}_i^{k-1} = \texttt{CrossAtt}(p_i^{k-1}, \mathcal{M}_{i,k}^t) \tag{10}$$

where $\tilde{p}_i^{k-1}$ denotes the memory-augmented query feature in decoder block $k$. After this, we further interact $\tilde{p}_i^{k-1}$ with multimodal motion and textual features, as follows,

$$p_i^k = \texttt{CrossAtt}(\tilde{p}_i^{k-1}, [\bar{f}_i^m, \tilde{f}_i^t]) \tag{11}$$

where $p_i^k$ is the learned query feature, and will be fed to next decoder block for further query feature learning. Fig. 6 shows our temporal memory selection can segment the video into different events and select the memory of the event closest to current moment, benefiting localization of target event. After $K$ blocks in the decoder, the temporal query feature $p_i^K$ is adopted for temporal prediction.

### 3.5 OPTIMIZATION

In ART-STVG, we predict both spatial bounding boxes and temporal start and end timestamps for loss computation. Due to limited space, please see our loss function in ***supplementary material***.

## 4 EXPERIMENTS

**Implementation.** ART-STVG is implemented with PyTorch (Paszke et al., 2019). We use ResNet-101 (He et al., 2016), VidSwin-tiny (Liu et al., 2022), and RoBERTa-base (Liu et al., 2019) for appearance, motion, and textual feature extraction. Following previous work (Gu et al., 2024; Jin et al., 2022a), we use pre-trained MDETR (Kamath et al., 2021) to initialize appearance and text backbones and multimodal fusion module. The hidden dimension of the encoder and decoder is $C = 256$, with channel dimensions of $C_a = 2048$, $C_m = 768$, and $C_t = 768$ for appearance, motion, and textual features. We sample video frames at FPS of 3.2 and resize each frame to have a short side of 420. The video frame length during training is $N_f = 64$, and the text sequence length is $N_t = 30$. During training, we adopt Adam (Kingma & Ba, 2015) with an initial learning rate of $1e-5$ for the pre-trained backbone and $1e-4$ for other modules, while keeping the motion backbone frozen.

**Datasets.** Since there are no benchmarks dedicated to LF-STVG, we opt to extend HCSTVG-v2 (Tang et al., 2021) for creating new datasets for LF-STVG. The ***reason*** for choosing HCSTVG-v2 only

for extension is that it is the only dataset which provides available source videos, thus allowing for extension with longer videos. Specifically, HCSTVG-v2 originally contains 16,000 video-sentence pairs in complex multi-person scenes, including 10,131 training, 2,000 validation, and 4,413 testing samples. Each video lasts 20 seconds and is paired with a textual query averaging 17.25 words. As annotations of the test set are not publicly available, the results are reported on the validation set, as in other methods (Yang et al., 2022; Lin et al., 2023b; Gu et al., 2024). For this reason, we extend only the validation set to lengths of 1 to 5 minutes, referred to as LF-STVG-1min/2min/3min/4min/5min, for the evaluation of LF-STVG. The extensions are based on original YouTube videos, not concatenated clips, and we manually review the extended videos to ensure their quality.

**Metrics.** Follow (Lin et al., 2023b; Jin et al., 2022a), we use m_tIoU, m_vIoU, and vIoU@R for evaluation. m_tIoU evaluates effectiveness of temporal grounding by averaging tIoU scores over all test videos. m_vIoU assesses spatial grounding performance by averaging vIoU scores. Additionally, vIoU@R measures performance by determining the proportion of test samples with vIoU scores exceeding a threshold R. For details, please see previous works (Lin et al., 2023b; Jin et al., 2022a).

## 4.1 Comparison on Long-Form STVG

To validate the effectiveness of ART-STVG on LF-STVG, we compare it to other methods on extended LF-STVG datasets. Please ***note***, all methods including ART-STVG are trained *exclusively* on the HCSTVG-v2 training set (average video length 20 seconds) for fair comparison.

Tab. 1 reports the results. As displayed in Tab. 1, our method significantly outperforms existing STVG methods in all metrics on all five datasets, showing the superiority of our ART-STVG in grounding target in long videos compared to existing models. Specifically, our method outperforms TA-STVG by achieving improvements in m_tIoU and m_vIoU of 0.7%/0.9%, 6.5%/5.1%, 9.1%/6.8%, 6.2%/4.9%, and 7.3%/5.5% scores across five different video lengths, respectively. In addition, compared with the baseline, which has a similar architecture to our ART-STVG but ***without*** memory and memory selection modules (please kindly check its architecture in ***supplementary material*** due to limited space), ART-STVG shows remarkable improvements on all the metrics under different video lengths as shown in Tab. 1, which demonstrates the importance of selective memories for LF-STVG.

Table 1: Comparison to other approaches on long-term videos. Our method shows the best results.

| Methods | m_tIoU | m_vIoU | vIoU@0.3 | vIoU@0.5 |
|---|---|---|---|---|
| *(a) LF-STVG-1min* | | | | |
| TubeDETR (Yang et al., 2022) | 32.5 | 20.8 | 25.7 | 8.7 |
| STCAT (Jin et al., 2022a) | 36.1 | 23.2 | 34.4 | 10.4 |
| CG-STVG (Gu et al., 2024) | 37.2 | 24.3 | 32.6 | 10.9 |
| TA-STVG (Gu et al., 2025) | 38.4 | 25.2 | 35.5 | 12.1 |
| Baseline (ours) | 30.1 | 19.7 | 25.5 | 8.3 |
| ART-STVG (ours) | **39.1** (+9.0) | **26.1** (+6.4) | **36.8** (+11.3) | **17.6** (+9.3) |
| *(b) LF-STVG-2min* | | | | |
| TubeDETR (Yang et al., 2022) | 23.0 | 13.4 | 10.9 | 2.5 |
| STCAT (Jin et al., 2022a) | 24.3 | 15.0 | 12.5 | 2.6 |
| CG-STVG (Gu et al., 2024) | 24.9 | 15.8 | 14.7 | 2.9 |
| TA-STVG (Gu et al., 2025) | 25.3 | 16.2 | 15.8 | 4.0 |
| Baseline (ours) | 23.0 | 15.1 | 16.5 | 6.6 |
| ART-STVG (ours) | **31.8** (+8.8) | **21.3** (+6.2) | **29.3** (+12.8) | **13.2** (+6.6) |
| *(c) LF-STVG-3min* | | | | |
| TubeDETR (Yang et al., 2022) | 13.6 | 6.4 | 7.2 | 2.9 |
| STCAT (Jin et al., 2022a) | 14.2 | 8.4 | 3.0 | 0.1 |
| CG-STVG (Gu et al., 2024) | 14.2 | 8.7 | 3.2 | 0.3 |
| TA-STVG (Gu et al., 2025) | 13.9 | 8.5 | 3.3 | 0.2 |
| Baseline (ours) | 16.2 | 10.7 | 10.5 | 4.5 |
| ART-STVG (ours) | **23.0** (+6.8) | **15.3** (+4.6) | **20.1** (+9.6) | **9.5** (+5.0) |
| *(d) LF-STVG-4min* | | | | |
| TubeDETR (Yang et al., 2022) | 9.6 | 5.2 | 1.2 | 0.1 |
| STCAT (Jin et al., 2022a) | 10.4 | 6.0 | 0.8 | 0.0 |
| CG-STVG (Gu et al., 2024) | 10.6 | 6.3 | 1.1 | 0.0 |
| TA-STVG (Gu et al., 2025) | 10.1 | 6.1 | 0.9 | 0.0 |
| Baseline (ours) | 9.9 | 6.2 | 4.7 | 1.4 |
| ART-STVG (ours) | **16.3** (+6.4) | **11.0** (+4.8) | **12.9** (+8.2) | **5.2** (+3.8) |
| *(e) LF-STVG-5min* | | | | |
| TubeDETR (Yang et al., 2022) | 7.8 | 3.9 | 0.7 | 0.1 |
| STCAT (Jin et al., 2022a) | 7.8 | 4.4 | 0.3 | 0.0 |
| CG-STVG (Gu et al., 2024) | 8.1 | 4.7 | 0.3 | 0.0 |
| TA-STVG (Gu et al., 2025) | 7.7 | 4.5 | 0.3 | 0.0 |
| Baseline (ours) | 9.2 | 5.3 | 4.5 | 1.1 |
| ART-STVG (ours) | **15.0** (+5.8) | **10.0** (+4.7) | **11.4** (+6.9) | **4.7** (+3.6) |

## 4.2 Ablation Study

To better understand our ART-STVG, we conduct extensive ablations on LF-STVG-3min.

**Impact of selective temporal memory.** We set up a temporal memory bank in temporal decoder to store target event information and use this temporal memory for locating start and end of event related to target. To verify its effectiveness, we conduct an ablation in Tab. 2. As in Tab. 2, without temporal memory, our method achieves an m_tIoU score of 16.7% (❶). When using all temporal memories, the m_tIoU score is decreased to 9.6% (❶ *v.s.* ❷). This is because the long-term video often contains multiple events, and using all temporal memories may introduce irrelevant information. When using our memory selection, the m_tIoU score is improved to 23.0% with 13.4% gains (❷ *v.s.* ❸). These results show our selective temporal memory can effectively improve ART-STVG for LF-STVG.

**Impact of selective spatial memory.** Similar to temporal decoder, we adopt a spatial memory bank in spatial decoder to learn contextual target information for spatial localization. We conduct an ablation

Table 2: Ablations of selective temporal memory.

| Temporal Decoder Memory | Selection | m_tIoU | m_vIoT | vIoU@0.3 | vIoU@0.5 |
|---|---|---|---|---|---|
| ❶ - | - | 16.7 | 11.1 | 11.9 | 4.7 |
| ❷ ✓ | - | 9.6 | 6.2 | 4.7 | 1.5 |
| ❸ ✓ | ✓ | **23.0** | **15.3** | **20.1** | **9.5** |

Table 3: Ablations of selective spatial memory.

| Spatial Decoder Memory | Selection | m_tIoU | m_vIoT | vIoU@0.3 | vIoU@0.5 |
|---|---|---|---|---|---|
| ❶ - | - | 21.3 | 13.9 | 16.4 | 8.0 |
| ❷ ✓ | - | 22.1 | 14.2 | 17.0 | 9.0 |
| ❸ ✓ | ✓ | **23.0** | **15.3** | **20.1** | **9.5** |

Table 4: Ablations of different decoder designs.

| Design Choice | m_tIoU | m_vIoU | vIoU@0.3 | vIoU@0.5 |
|---|---|---|---|---|
| ❶ Parallel | 21.5 | 13.9 | 17.3 | 8.2 |
| ❷ Cascaded (ours) | **23.0** | **15.3** | **20.1** | **9.5** |

Table 5: Ablations of different choices for $N_s$.

| | m_tIoU | m_vIoU | vIoU@0.3 | vIoU@0.5 |
|---|---|---|---|---|
| ❶ $N_s = 16$ | 22.7 | 15.0 | 18.4 | 9.2 |
| ❷ $N_s = 32$ (ours) | **23.0** | **15.3** | **20.1** | **9.5** |
| ❸ $N_s = 48$ | 22.5 | 14.7 | 18.2 | 9.1 |

Table 6: Ablations of training with longer videos. Please notice that, all the compared approaches are trained on the 40-second videos using their provided source codes for fair comparison.

| Methods | m_tIoU | m_vIoU | vIoU@0.3 | vIoU@0.5 |
|---|---|---|---|---|
| TubeDETR (Yang et al., 2022) | 20.8 | 11.5 | 9.8 | 3.9 |
| STCAT (Jin et al., 2022a) | 21.0 | 12.2 | 7.4 | 0.6 |
| CG-STVG (Gu et al., 2024) | 20.5 | 12.0 | 8.0 | 1.0 |
| TA-STVG (Gu et al., 2025) | 20.7 | 11.8 | 7.7 | 0.5 |
| ART-STVG (ours) | **28.3** | **18.8** | **27.0** | **11.9** |

in Tab. 3. We observe that integrating all spatial memories can improve the m_tIoU score to 22.1% with 0.8% gains (❶ *v.s.* ❷), and applying the memory selection strategy can further enhance the m_tIoU score to 23.0% with 0.9% gains (❷ *v.s.* ❸), validating the importance of selective memory.

**Impact of design for spatial and temporal decoders.** We introduce a cascaded spatio-temporal design in ART-STVG, which allows the use of fine-grained target information from spatial grounding to assist temporal localization in complex long videos. To validate its efficacy, we conduct an ablation in Tab. 4. From Tab. 4, it is evident that cascading spatial and temporal decoders outperforms the parallel design with improvements of 1.5% and 1.4% scores on m_tIou and m_vIoU (❶ *v.s.* ❷).

**Impact of the number of selective spatial memories.** In the spatial decoder, we utilize $N_s$ to control the number of selective spatial memories. To explore the impact of $N_s$, we conducted the ablation experiment in Tab. 5. We can see that when $N_s$ is 32, the performance of the model is the best (❷).

**Impact of the length of training videos.** To investigate the impact of training videos of different lengths, we extend HCSTVG-v2 training set to 40 seconds and use it to train both existing methods and ART-STVG. As in Tab. 6, we can see all methods show clear gains when trained on 40-second videos compared to 20-second videos (Tab. 6 *v.s.* Tab. 1 (c)). This shows that training with longer videos enhances target localization in long-term videos, yet results in increasing training costs. More importantly, our method still achieves the best performance on all metrics.

### 4.3 COMPARISON ON SHORT-FORM STVG

We further evaluate ART-STVG on SF-STVG in Tab. 7 on HCSTVG-v2 validation set. As in Tab. 7, our method shows competitive results to current STVG methods on short-term videos. Current methods use non-autoregressive structures that process video frames in parallel to capture inter-frame relationships, and are specially designed for target localization in short-term videos. Despite this, ART-STVG, adopting an autoregressive structure, outperforms most existing methods, falling only behind TA-STVG (Gu et al., 2025) by 1.2%/1.0%

Table 7: Comparison on SF-STVG.

| Methods | m_tIoU | m_vIoU |
|---|---|---|
| 2D-Tan (Tan et al., 2021) | - | 30.4 |
| MMN (Wang et al., 2022) | - | 30.3 |
| TubeDETR (Yang et al., 2022) | 53.9 | 36.4 |
| STCAT (Jin et al., 2022a) | 56.6 | 36.9 |
| STVGFormer (Lin et al., 2023b) | 58.1 | 38.7 |
| CG-STVG (Gu et al., 2024) | 60.0 | 39.5 |
| TA-STVG (Gu et al., 2025) | **60.4** | **40.2** |
| Baseline (ours) | 46.2 | 29.9 |
| ART-STVG (ours) | 59.2 | 39.2 |

in m_tIoU/m_vIoU. Moreover, our method shows clear gains compared to baseline without memory.

Due to limited space, we show additional results, analysis, and discussions in ***supplementary material***.

## 5 CONCLUSION

In this work, we study Long-Form STVG, and propose a new framework, ART-STVG, that can handle long-term videos effectively. The core of ART-STVG lies in the use of selective memories, which are applied to decoders for leveraging spatio-temporal contextual cues for grounding, greatly improving performance. Additionally, our cascaded spatio-temporal decoder design effectively exploits spatial localization to assist temporal localization in long-term videos. On multiple extended LF-STVG datasets, ART-STVG significantly outperforms other methods, showing its superiority.

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

## SUPPLEMENTARY MATERIAL

For a better understanding of this work, we offer additional details, analysis, and results as follows:

- **A** *Details of Optimization*
  In this section, we provide more details of our loss function for optimization.
- **B** *Design of Baseline*
  In this section, we introduce the architecture of the baseline method.
- **C** *Analysis of Efficiency*
  In this section, we analyze the efficiency and complexity of ART-STVG and compare it to other state-of-the-art methods.
- **D** *Analysis of Failure Cases*
  In this section, we discuss failure cases of our proposed method.
- **E** *Analysis of Qualitative Results*
  In this section, we show qualitative results of our method on LF-STVG and comparison to the baseline method.
- **F** *Comparison with Existing Memory-based Video Understanding Works*
  In this section, we discuss differences with existing memory-based video understanding methods.
- **G** *Limitation and Broader Impact*
  In this section, we discuss the limitation of our method and its broader impact.

## A DETAILS OF OPTIMIZATION

Given the video containing $N_f$ frames and its textual query, our ART-STVG predicts: (1) object boxes $\mathcal{B} = \{b_i\}_{i=1}^{N_f}$ in the memory-augmented spatial decoder; (2) event start timestamps $\mathcal{H}_s = \{h_i^s\}_{i=1}^{N_f}$ and end timestamps $\mathcal{H}_e = \{h_i^e\}_{i=1}^{N_f}$ in the memory-augmented temporal decoder. During the training, with the groundtruth of the bounding box $\mathcal{B}^*$, start timestamps $\mathcal{H}_s^*$ and end timestamps $\mathcal{H}_e^*$, we can calculate the total loss $\mathcal{L}$ as

$$\mathcal{L} = \underbrace{\lambda_k(\mathcal{L}_{\text{KL}}(\mathcal{H}_s^*, \mathcal{H}_s) + \mathcal{L}_{\text{KL}}(\mathcal{H}_e^*, \mathcal{H}_e))}_{\text{loss of memory-augmented temporal decoder}} + \underbrace{\lambda_l \mathcal{L}_1(\mathcal{B}^*, \mathcal{B}) + \lambda_u \mathcal{L}_{\text{IoU}}(\mathcal{B}^*, \mathcal{B})}_{\text{loss of memory-augmented spatial decoder}} \tag{12}$$

where $\mathcal{L}_{\text{KL}}$, $\mathcal{L}_1$ and $\mathcal{L}_{\text{IoU}}$ are KL divergence, smooth L1 and IoU losses. $\lambda_k$, $\lambda_l$ and $\lambda_u$ are parameters to balance the loss. Similar to previous methods (Jin et al., 2022b; Lin et al., 2023b; Gu et al., 2024), $\lambda_k$, $\lambda_l$ and $\lambda_u$ are empirically set to 10, 5, and 3, respectively.

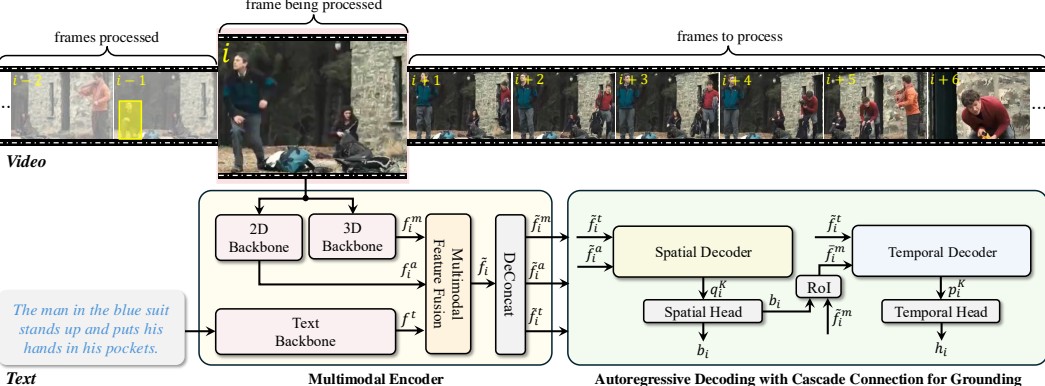

Figure 7: Architecture of baseline, containing an encoder and an autoregressive decoder but *without* memories and memory-related modules.

## B    DESIGN OF BASELINE

The baseline method mentioned in the main text shares a similar architecture with ART-STVG but does *not* contain the (spatial and temporal) memories and memory-related modules. Its detailed architecture is shown in Fig. 7. Specifically, the baseline framework comprises two core components: a multimodal encoder for extracting and fusing cross-modal features, followed by cascaded spatial and temporal decoders that perform autoregressive decoding to progressively localize the target. Such architecture makes it suitable for long video processing. However, without memory, its performance is inferior to our ART-STVG (please see the comparison of our ART-STVG and the baseline in the main text), which evidences the necessity and importance of our memory design for improving LF-STVG performance.

## C    ANALYSIS OF EFFICIENCY

To analyze the efficacy and complexity of our model, we report the efficiency of the model and the comparison with other methods in Tab. 8. As shown in Tab. 8, our model size is similar to other methods. Although our inference time (for 64 images, aligned to SF-STVG methods) is longer due to autoregressive processing, at 1.09 seconds, compared to the inference times of STCAT, CG-STVG, and TA-STVG, which are 0.47, 0.71, and 0.69 seconds respectively, our GPU memory usage is much lower than other methods. Specifically, our GPU memory usage is 7.9G, while other methods such as TA-STVG and CG-STVG have GPU memory usages of 25.1G and 25.9G, respectively. Therefore, our method is more suitable for handling long videos.

Table 8: Comparison of model efficacy and complexity on a single A100 GPU.

| Methods | Model Size | | Inference | |
| --- | --- | --- | --- | --- |
| | Trainable Params | Total Params | Time | GPU Memory |
| STCAT (Jin et al., 2022b) [NeurIPS'2022] | 207 M | 207 M | 0.47 s | 23.6 G |
| CG-STVG (Gu et al., 2024) [CVPR'2024] | 203 M | 231 M | 0.71 s | 25.9 G |
| TA-STVG (Gu et al., 2025) [ICLR'2025] | 206 M | 234 M | 0.69 s | 25.1 G |
| ART-STVG (ours) | 207 M | 235 M | 1.09 s | 7.9 G |

## D    ANALYSIS OF FAILURE CASES

Despite promising performance on LF-STVG, our method may fail in some complex scenes: *(i) indistinct event boundaries*. Detection of event boundaries in long videos is crucial for temporal memory selection in ART-STVG. When event boundaries are ambiguous in videos, their detection may be compromised, which results in inaccurate temporal memory selection and hence degrades temporal localization in STVG; for example, in Fig. 8-top, the start time of the target event is ambiguous. *(ii) highly distracting background objects*. When there exist highly distracting background targets (that have similar appearance and action with the foreground target) in videos, our method may drift to background targets, leading to spatial localization failure. In Fig. 8-middle, there are two people with similar actions. *(iii) Extremely short target events.* When the target event lasts for a very short duration within a long video, the presence of a large amount of redundant information makes grounding more difficult. For example, in Fig. 8-bottom, the target event lasts only 3 seconds, while the entire video is

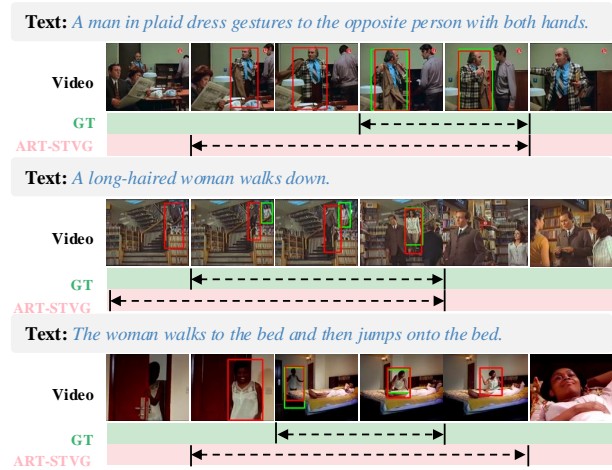

Figure 8: Failure cases. The red box in the figures indicates the results of ART-STVG, while the green box indicates the ground truth (GT).

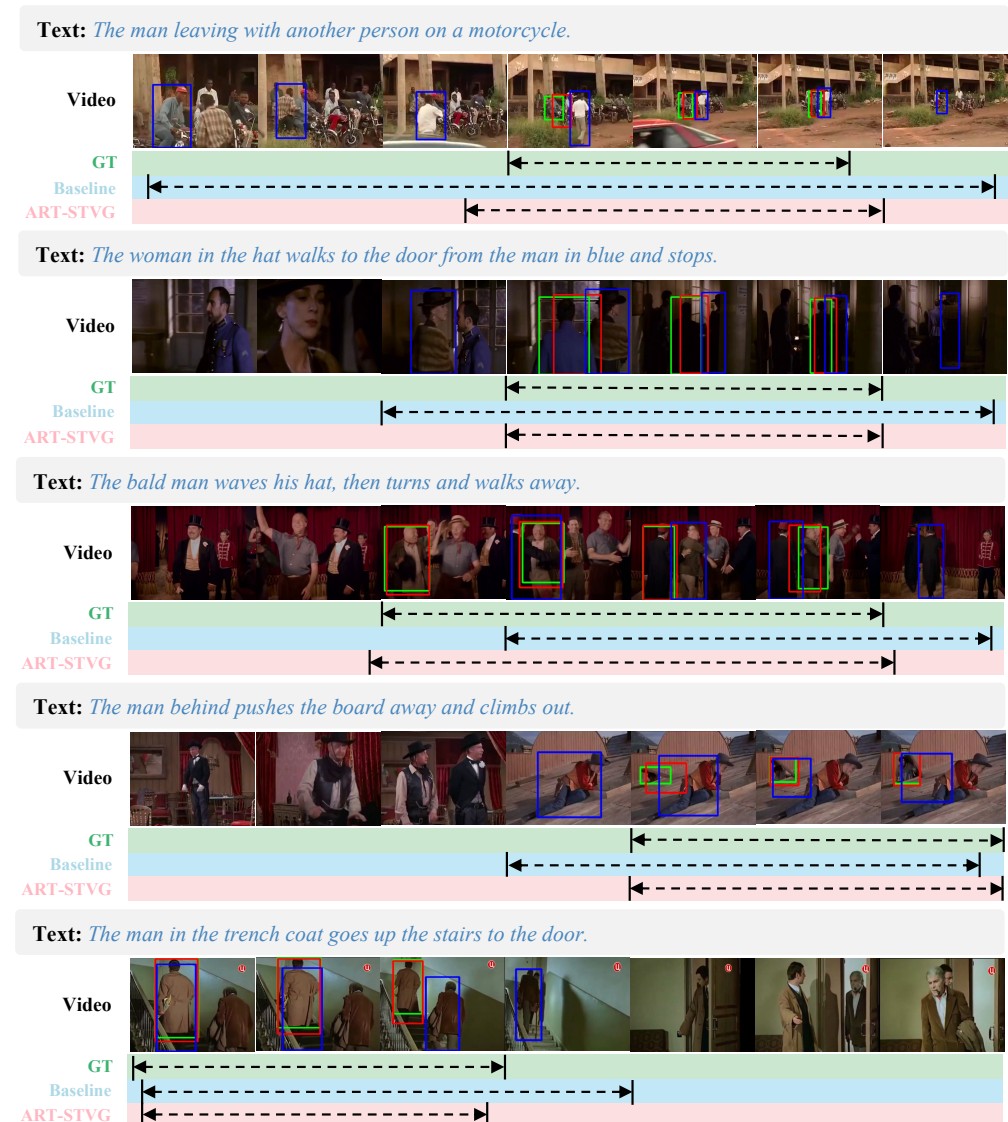

Figure 9: Qualitative results of ART-STVG, the Baseline without memory, and the Ground Truth.

300 seconds long, making localization hard. To handle the above cases, we will explore fine-grained target and event cues in videos for improvements in future work.

## E    ANALYSIS OF QUALITATIVE RESULTS

To qualitatively validate our method on LF-STVG, we present grounding results and comparisons with the baseline method without memory on the LF-STVG-3min benchmark. From Fig. 9, we observe that the baseline without memory often locates inconsistent targets across different video frames, such as in the third and fifth examples. In contrast, our method locates the target more consistently and accurately, demonstrating the effectiveness of our approach.

## F    COMPARISON WITH MEMORY-BASED VIDEO UNDERSTANDING WORKS

The memory bank has been explored in long video understanding (He et al., 2024; Song et al., 2024; Qian et al., 2025). However, our proposed memory bank in ART-STVG is *different* than these

approaches in two aspects: *(1) Different memory information*. The memory in (He et al., 2024; Song et al., 2024; Qian et al., 2025) is global context information for the entire video, while memory in our method is *text-guided spatial instance and temporal event boundary cues*, specially designed for LF-STVG; *(2) Different memory compression for selection*. For memory selection, our method merges spatial memories using *text as guidance* and temporal memories using *event boundary cues from videos*, with both specially designed for LF-STVG, while the mentioned works do not have these mechanisms as they aim at different tasks.

## G  LIMITATION AND BROADER IMPACT

**Discussion of Limitation.** As the first work to explore the LF-STVG problem, our ART-STVG shows promising performance on long-term videos and significantly outperforms existing STVG methods. Despite this, our method has two limitations. First, although ART-STVG is capable of handling long videos, yet its performance may degrade as the video becomes longer and more complex. To mitigate this issue, a possible direction is to learn more discriminative memory systems for capturing fine-grained target information. Since this is beyond our current goal of attempting the LF-STVG problem, we leave it to our future work for improving LF-STVG. Second, similar to other approaches, our method cannot operate in real-time, which may limit its applications in certain scenarios. In the future, we will study lightweight architectures for LF-STVG using techniques from model compression and quantization.

**Discussion of Broader Impact.** The proposed ART-STVG focuses on localizing the target of interest within an untrimmed video given a textual description. This technique has a wide range of crucial applications, including content-based video retrieval, video content moderation, and sports analytics. By developing ART-STVG, we expect it to contribute positively to societal advancements and technological progress.

