# OpenReview forum: "Towards Long-Form Spatio-Temporal Video Grounding"
_ICLR.cc/2026/Conference — ICLR 2026 Conference Withdrawn Submission_

### Official Review · Reviewer_wm7F · 2025-10-16

**Soundness:** 3
**Presentation:** 3
**Contribution:** 2
**Rating:** 4
**Confidence:** 4

**Summary:**

This paper first introduces a new task, i.e., long-form spatio-temporal video grounding (LF-STVG), and proposes the ART-STVG, an autoregressive, memory-augmented transformer for LF-STVG. It processes video frame-by-frame, maintains spatial and temporal memory banks with selective retrieval, and uses a cascaded design so temporal localization benefits from the spatial prediction. The authors also extend HCSTVG-v2 to 1–5-minute validation and report large gains over TubeDETR, STCAT, CG-STVG, and TA-STVG on these long-form sets.

**Strengths:**

- This paper introduces a new task setting, i.e., long-form spatio-temporal video grounding (LF-STVG).
- The authors also extend HCSTVG-v2 to 1–5-minute validation and report large gains over TubeDETR, STCAT, CG-STVG, and TA-STVG on these long-form sets.
- This paper is easy to understand.

**Weaknesses:**

- Although the paper frames LF-STVG as long-form, the maximum evaluated duration (~5 minutes) is modest when compared to datasets like MAD, where the average video spans ~110.8 minutes [a]. Moreover, OmniSTVG [b] investigates long-form conditions and multi-object detection, thereby encompassing a more comprehensive setting.
- ART-STVG seems to assemble known techniques (self-attention fusion, memory banks, selection mechanisms), and the specific novelty remains unclear.
- This paper claims that previous methods generally process all frames in one time, so it treats the video as a streaming input and processes its frames sequentially. However, in feature extraction, VideSwin is used to extract the motion features with previous frames as input. That means when processing the last frame, all frames will be used for the motion feature extraction. And another question is each time the current frame is processed, are all the previous frames used again? Wouldn’t that increase the computational load? I’d like to know how the training and inference time of this method compares with previous methods.
- In this paper, FPS is set to 3.2. How will it influence the performance?

[a] Mad: A scalable dataset for language grounding in videos from movie audio descriptions, CVPR 2022.
[b] OmniSTVG: Toward Spatio-Temporal Omni-Object Video Grounding, arxiv 2025.

**Questions:**

See the detailed questions in weaknesses.

---

### Official Review · Reviewer_c8Pk · 2025-10-21

**Soundness:** 3
**Presentation:** 3
**Contribution:** 3
**Rating:** 6
**Confidence:** 5

**Summary:**

The paper first studies Long-Form Spatio-Temporal Video Grounding (LF-STVG) and proposes a new framework ART-STVG that handles long-term videos. It treats the video as a streaming input and processes its frames sequentially. It proposes spatial and temporal memory banks to capture spatio-temporal context and designs memory selective strategies that enable more relevant information for the decoders. Rather than parallelizing spatial and temporal localization, it introduces a cascaded spatio-temporal design. On the newly extended datasets for LF-STVG, it outperforms current Short-Form STVG methods.

**Strengths:**

1.	It is well-motivated to be the first to explore the LF-STVG problem and propose the first framework attempting to handle LF-STVG.
2.	It achieves SOTA results on extended datasets for LF-STVG, while maintains competitive results on SF-STVG.
3.	The writing is clear, and the presentation of figures and diagrams is good.

**Weaknesses:**

1.	Some ablation studies are missing. For example, what is the impact of the number of selective temporal memories? What is the impact of the order of spatial decoder and temporal decoder?
2.	More detailed discussion should be complemented. Since ART-STVG trades time for space by ingesting frames one at a time, detailed discussion on the trade-off between space and time is valuable to verify the balance in practical scenes.
3.	More explanation should be given. Generally, longer videos bring richer spatiotemporal information and better performance. However, it is counter-intuitive that the performance gain decreases from LF-STVG-1min to LF-STVG-5min in Table 1. Therefore, more explanations or discussions about this phenomenon are needed.

**Questions:**

1.	What is the performance when using different 2D backbone, 3D backbone, and text backbone?
2.	What is the performance of directly splitting long-term videos into multiple short clips, and then using current SF-STVG methods to do STVG?
3.	What is the performance of ART-STVG compared to larger models, like Multimodal LLMs used for spatio-temporal video grounding (e.g., [a])?

[a] Unleashing the Potential of Multimodal LLMs for Zero-Shot Spatio-Temporal Video Grounding, arXiv 2025.

---

### Official Review · Reviewer_He6Z · 2025-11-01

**Soundness:** 2
**Presentation:** 3
**Contribution:** 2
**Rating:** 4
**Confidence:** 4

**Summary:**

This paper proposes ART-STVG, a memory-augmented framework for Long-Form STVG. It models long-term dependencies via selective memory and employs a cascaded decoder to leverage spatial cues for temporal grounding. ART-STVG significantly outperforms existing methods on extended datasets of HCSTVG-v2 and demonstrates strong generalization to short videos.

**Strengths:**

1.	This paper introduces ART-STVG for long-form spatio-temporal video grounding (LF-STVG), using frame-wise autoregressive decoding to mitigate the memory burden and irrelevant distractions of parallel full-clip processing.
2.	It proposes novel spatio-temporal memory selection strategies: spatially, it selects context by comparing the text query with memory slots; temporally, it segments by comparing adjacent memory slots and retrieves the most recent segment—an effective design for long-video scenarios.
3.	It adopts a well-motivated spatial→temporal cascade, leveraging fine-grained spatial evidence to more accurately localize temporal boundaries.
4.	On HC-STVG-v2, the method delivers consistent gains and ablations verify the independent contributions of each component.

**Weaknesses:**

1.	All experiments are conducted on the extended HC-STVG-v2 long-video validation set, with no evaluation on other grounding benchmarks such as VidSTG [A] (short/long). The model’s generalization therefore remains to be further verified.
2.	The structural novelty is moderate. The adopted mechanisms—multimodal feature extraction and fusion, selective memory, and memory banks—have been widely used in video QA, temporal localization, and long-video modeling. The paper’s main contribution lies in task-oriented integration and pruning for fine-grained STVG via an autoregressive streaming pipeline plus a spatial→temporal cascade with text-guided memory selection.
3.	How does the size of the memory bank grow with video length? Is there an eviction/compression policy, and what is its quantitative impact on latency and memory usage?
4.	Autoregression reduces memory but increases latency; the main paper lacks scaling curves of latency/memory versus video length, so the boundary of the engineering trade-off is unclear.
5.	The memory selection relies on heuristic similarity rules (text–memory and adjacent-memory); comparisons against learnable or uncertainty-based alternatives are missing.

**Questions:**

See the comments above.

---

### Official Review · Reviewer_K7x2 · 2025-11-01

**Soundness:** 3
**Presentation:** 3
**Contribution:** 2
**Rating:** 4
**Confidence:** 5

**Summary:**

This work proposes a new benchmark for the new task of Long-Form Spatio-Temporal Video Grounding (LF-STVG) along with a new framework dubbed ART-STVG specifically designed for LF-STVG. LF-STVG extends the video duration of traditional short-form STVG from tens of seconds to several minutes and the ART-STVG is designed as a streaming video processing method incorporated by spatial and temporal memories to predict the spatio-temporal localization results frame by frame. Comparisons and experiments validate the superiority of the proposed benchmark and framework over prior ones.

**Strengths:**

1.The motivation of the proposed task and the model design is intuitively reasonable and technically sound.

2.LF-STVG is a practical and promising new direction that opens up new opportunities in localization-oriented multimodal video understanding area.

3.Extensive experiments verify the drawbacks of previous SF-STVG methods in handling long video scenarios and the proposed method provides a strong baseline in long-form STVG.

**Weaknesses:**

1.One of my biggest concerns regarding this work is the lack of the tailored training data for LF-STVG. As mentioned in the manuscript, the authors extended the validation set of a short-form STVG dataset HCSTVG-v2 to several minutes, and the experimental results are reported for models trained on existing short-form STVG data. This limitation decreases the contribution of this work as I think LF-STVG is a very challenging task and needs tailored training data to facilitate its development in the community.

2.As the authors claimed that they proposed a new streaming video processing paradigm for LF-STVG, it would be better to compare the proposed method with some streaming video understanding methods to make the paper's contributions clearer and more convincing.

3.There are some unclear expressions regarding the model details. For example, the shapes of the spatial and temporal memories are missing and the update mechanism of these memories also lack clear clarifications.

**Questions:**

Please refer to the weaknesses.

**Details Of Ethics Concerns:**

N/A.

---

### Note · Authors · 2025-11-15

I have read and agree with the venue's withdrawal policy on behalf of myself and my co-authors.